# Proposed standards for prosthetic foot reuse and considerations for donation of used prosthetic feet to low-and middle-income countries

Michael A. Berthaume[1]*, Louise Ackers[2], Laurence Kenney[3], Vikranth Harthikote Nagaraja[3], Promise Maduako[4]

1 Department of Engineering, King's College London, London, United Kingdom, 2 School of Health and Society, University of Salford, Salford, United Kingdom, 3 Centre for Human Movement and Rehabilitation Research, University of Salford, Salford, United Kingdom, 4 STAND, Backfields House, Bristol, United Kingdom

* michael.berthaume@kcl.ac.uk

## Abstract

Prosthetic components from high-income countries (HICs) are often replaced not because they are broken, but because of guidelines or expired warranties, meaning they may still be usable. As most HICs classify prostheses as single patient multi-use devices, components are often disposed of or donated to low- and middle- income countries (LMICs) where medical device regulatory frameworks are limited or non-existent. A lack of standards guaranteeing the quality of donated prosthetic components could lead to a violation of the World Health Organization's principles of good donation. Here, we work towards the creation of a set of standards by quantifying the efficacy of a second-hand donated prosthetic foot quality checklist developed by STAND. We compared 170 checked to 196 unchecked feet received by prosthetic and orthotic centres in Fort Portal, Uganda, and found checklist implementation increased the percentage of usable feet from 83.16% to 94.12%. Foot brand significantly affected usability, but further data and samples are needed to disentangle the effects of prosthetic foot brand, prosthetic foot model, and centre from which the feet originated on prosthetic foot usability. We propose a rapid and efficient quality assurance checklist as a first step towards a set of standards towards prosthetic foot reuse and discuss future research directions. Research towards the creation of an international set of standards/regulatory requirements governing the use of prosthetic limbs, like the international standards used for prosthetic limb design, would not only enable the safe, useful provision of prosthetic components in LMICs, but would also set the groundwork for understanding how a repair, reuse, and recycle model for prosthetic components might be implemented in HICs. Globally, this could decrease prosthetic provision time, create a circular economy for prosthetic components, and reduce the carbon footprint of prosthetic component manufacture and provision.

**Data availability statement:** All relevant data are within the paper and Supporting information files.

**Funding:** This work was funded by Knowledge for Change (K4C), which LA is the Director of, and has no grant number. The funders had no role in study design. They aided in data collection, interpretation of the results, decision to publish, or preparation of the manuscript.

**Competing interests:** PM works for STAND, who donated the prosthetic feet used in this study. LA is a founder of Knowledge for Change, which aids in the supply of the prosthetic feet in Uganda, and helped establish prosthetic and orthotic services at the Ninsiima Centre for the Rehabilitation of People with Physical Disabilities. The feet from K4C were also stored at the Knowledge For Change offices. There are no patents associated with this research to declare. The usability of several market products are associated with this research. Namely, prosthetic feet from the following companies: Janton, Steeper, Fillauer, Kingsley Casual, BioQuest, Blatchfords, College Park Industries (CPI), Ottobock, WillowWood, Trulife, Össur, and Streifender. These products were not donated by the listed companies, and the companies have no involvement in this research. These competing interests do not alter our adherence to PLOS Global Public Health policies on sharing data and materials.

## Introduction

Despite being a priority assistive product and a basic human right (Convention on the Rights of Persons with Disabilities; [1]) and critical for the sustainable development goals (SDG 3: Good Health and Well-Being), poor levels of access to prosthetic limbs are common in low- and low-and middle-income countries (LICs, LMICs). Systematic and systemic issues arise in terms of funds (e.g., to purchase prosthetic devices, or pay for repairs), access to professionals (prosthetists, technicians, etc.) and materials (e.g., polypropylene), and access to prosthetic and orthotic (P&O) centres by the patients, which are exacerbated in rural areas [2]. Various methods of prosthetic provision have arisen to meet these challenges.

### Prosthetic provision in LMICs

Prosthetic provision in LMICs involves inter- and intra-national stakeholders. This is particularly true during times of conflict, when individuals with limb loss are a visible reminder of the effects of conflict on the person, and the media and public draw a moral imperative, empowering many stakeholders to help [3]. Stakeholders include non-profit organizations (NGOs), charities and faith-based organizations (FBOs), for-profit and hybrid for-profit/NGOs, prosthetic training centres, and hospitals/P&O centres themselves [2]. This creates a mostly "top-down" approach from high-income countries (HICs) to LMICs [4], but bottom-up approaches, such as Community-Based Rehabilitation (CBR), exist, and can be integrated into hybrid models (e.g., NGOs that participate in CBR). CBR utilizes local social and community infrastructure and provides some benefits (Cummings, 1996), particularly in creating limb-loss support networks [5].Provision of prosthetic devices includes prosthetic hardware, (locally) manufactured (bespoke) sockets, and the surrounding systems and services, such as local skillsets and supply chains [6,7]. When providing prostheses, cultural, economic, social, psychological, religious, and climatic factors, as well as the integration into local systems, should impact the choice of technology/services provided [4]. Poonekar suggested LMIC prostheses should be (i) low-cost; (ii) locally available; (iii) capable of manual fabrication; (iv) considerate of local climate and working conditions; (v) durable; (vi) simple to repair; (vii) simple to process using local production capability; (viii) reproducible by local personnel; (ix) technically functional; (x) biomechanically appropriate; (xi) as lightweight as possible; (xii) adequately cosmetic; and (xiii) psychosocially acceptable [8].

Cultural and sociocultural norms are often overlooked but can be particularly important. For example, karma may lead people to believe individuals lost limbs because of cosmic balance and can lead to an indifference in limb-loss aid, whereas the "alms for the poor" mentality can lead to larger community engagement and aid [3]. Heim (1979) discusses case studies where a lower-limb prosthetic user refused a 'peg-leg' because other users did not have peg-legs and they wanted to be treated as an equal. Similarly, when patellar tendon bearing (PTB) prostheses were being given out in a tribe, one man – who was not given a PTB socket – wanted one to be like the others [9]. Conversely, the introduction of new technologies can be difficult to

accept due to anxiety about the technology's performance or embarrassment about using the technology for the first time, particularly among first-time prosthetic users [10]. Acceptance can increase once users see proof new technologies work well with others [9]. If assistive devices fail to meet cultural needs, they can be abandoned [11–13], although the extent of device abandonment in LMIC settings is an under-researched area. "… unsuitability of assistive devices for the topographical and cultural (social and attitudinal) contexts in many developing countries, which are then strongly linked to poor patient outcomes, including dissatisfaction and limited participation in meaningful activities [14]." Within these contexts, it is important to remember culture is fluid and ever changing. The interdisciplinary approach combining anthropological (e.g., cultural) and engineering concepts can lead to the improved design and provision of prostheses [15–17].

Despite the best of intentions, prosthetic provision does not always go well [18], and the provision of inappropriate technologies or services can cause harm [5]. Ignorance of social/environmental issues, as well as grouping of all LICs/LMICs, developing countries, and/or low-resource settings into the same categories, is problematic. "The commonalities of poverty, lack of resources, demographics and the prevalence of diseases, which have been eradicated in the rest of the world, can lead to sweeping generalisations regarding the needs of LICs. Unique aspects of each country's history, politics, culture and social structure must be considered when planning any prosthetic and orthotic service [19]." Prosthetic providers may wish to provide a large range of products to a single end-user group to ensure all end-user needs are met. However, this approach may not be practical, as it requires sustained access to a broad range of supplies and skillsets [9].

There are, however, some common issues with prosthetic provision across all LMICs. Charity-based and globalisation models are the most prominent models of assistive technology provision in LMICs [6,20]. Bulk manufacturing can lead to a decrease in assistive technology cost, but inadequate buying power by providers in LMICs can prevent them from realizing these benefits [21]. Bulk production can also lead to a limited range of prosthetic technologies available [21] – rather than choosing the best prosthetic technology/service for a population, prosthetic technologies/services are presented based on availability to/by the organisation [3]. Even when choice is available, once in country, products are often provided on a first-come first-serve basis as materials are received instead of the most appropriate technologies being given to each person [5]. As globalization, which tends to focus on money spent or number of prostheses given and not how the user's life has improved [5,22], can disrupt local supply chains and squash local innovation [6,20], it can be difficult to develop and/or provide devices and services which fulfil end-user needs.

## The donation model of prosthetic provision

The donation model for prosthetic provision – which overlaps with the reuse and repair models [6,23] – is embedded within FBOs, NGOs, and for-profit/hybrid organizations [2], where it is the responsibility of the donor to ensure device quality and safety are met. It operates in a complex ecosystem which involves a diverse set of stakeholders, who often navigate numerous barriers affecting quality and appropriateness of donations, donation sustainability, and donation optimization [24].

If new or second-hand products meet standards and regulatory requirements, they can be a major supply source for some countries, providing advanced, otherwise unavailable technologies for cheaper-than-market prices [6,21]. While donated medical devices typically meet safety, quality, and performance requirements, Nasir et al. (2023) found regulatory and guidelines/processes for donated medical devices in African countries to be inadequate. This is partially due to oversight and ill-defined/poorly understood medical device regulations. Similar issues have been noted elsewhere (e.g., [25–27]).

Standards for reuse, repair, and donation of devices can exist at the organization, country, and international level. For example, an organization may only donate devices that reach certain internal quality standards, while governments and international organizations, like the World Health Organization (WHO), may set their own standards or regulatory requirements. One speaker at a recent meeting organised by the Ugandan Ministry of Health in support of the Ugandan Rehabilitation and Assistive Devices Strategic Plan asked foreign organisations to 'stop dumping their junk on us', highlighting the need for quality standards which are set and adhered to by international organizations. Several checklists, guidance, and

frameworks have been established to standardize donation of medical devices and equipment, highlighting the need to consider the donation process holistically [24,28].

When donating, it is good practice to adhere to WHO principles of good donation. However, it is important to recognize these are not international regulations, but guidelines meant to help national and institutional organizations deal with health care equipment donations, and organizations should create their own standards/regulations [29]. These include 1) ensuring health care equipment benefits the recipient to the maximum possible extent, 2) providing donations that are desired and fit local standards, 3) ensuring there are no double standards in quality, and 4) effective communication between donor and recipient while ensuring agreed upon donation plans are adhered to. Donators should work with local stakeholders and coordinate with national service systems, ensuring consistency of service, follow-up, data control, choice of technology, trained personnel, and sustainability [28,30,31]. The provision of product specifications and use guidelines in local languages aids with technological sustainability is the goal [30,32]. Only working/needed equipment should be donated, and recipients – be that countries, organizations, or device users – should be allowed to turn down donations. Time and budget should be allocated to follow-up on donations, e.g., for maintenance and repairs [32], and donators should stay involved for years for the establishment of prosthetic services, to ensure longevity and sustainability [9]. Achieving this requires genuine bilateral partnership and agreement on both sides for detailed accountability.

Despite distinct advantages with the donation model, issues exist with sustainability, quality, compatibility, infrastructure, and training [33], and the donation of medical equipment can unfortunately have adverse effects [32,34]. In the surgical/anaesthesiology sectors, "Lack of planning and collaboration can mean that equipment donated with 'good intentions' to help address these shortages is inappropriate, ineffective or dangerous [32]." Even when donations are high-quality and appropriate, differences in training and reliable infrastructure availability can mean technologies that work well in HICs work poorly/are unsustainable in LMICs [32]. Within assistive technologies, donated devices can fail users in the short-term by having inconsistent supplies, being of low quality, and/or being inappropriate for the context of use, e.g., not working well in the rugged environments of the target countries [2,6,21,30]. For example, "Donations from [overseas] of used hearing aids, they used us as a dumping ground, they were not functioning well [30]." In the long-term, failure can negatively affect quality of life in LMICs [14], and shifts responsibility for the disposal of medical devices to already overstretched local services. Medical device companies are not obliged to translate their documentation into languages of donation-recipient countries, and the lack of doing so is bad practice [30,32] and inevitably causes challenges, particularly with the more complex devices [35].

Donated items come with reduced lifetime and performance and increased servicing and maintenance challenges [12]. Products can be difficult to repair as it may not be possible to obtain replacement parts – putting reliability of future supply on the donors [20] and sometimes meaning recipient countries re-use single-use items – and/or personnel may not have proper training [32]. "Even when projects utilise donated western devices, their complexity predisposes then to require regular repair and replacement. It is difficult to refuse free devices even when they are not fit for purpose [19]." While donated products/services provide immediate relief (assuming they are not abandoned by the user [5]), they can lead to dependency on external sources, preventing the establishment of local supply chains and hindering the development of local manufacturing and maintenance capabilities [36]. Ultimately, good intentions are not enough; partnerships involving the use of donated componentry should consider human resources, environment, material resources, maintenance resources, and education resources when agreeing to provide and receive prosthetic components [32]. While many of these responsibilities fall on the donors, recipients should not be viewed or treated as passive, and partnership agreements should exist where receiving organisation(s) take some responsibility and do not agree to receive components unless they can manage them properly.

### Donation of prosthetic lower limbs and their componentry

There are no international standards and regulatory requirements governing the use of used prosthetic limbs or their componentry, and there is limited work on P&O donation in LMICs [37], meaning there is no way to guarantee prosthetic component quality or safety when reused. Consequently, many HICs/companies classify prostheses and prosthetic

components as single-patient multi-use medical devices which can be repaired and given back to the same patient when in warranty, and the inclusion of out-of-warranty used prosthetic components in new prostheses is not allowed. However, as prosthetic devices are regularly repaired and provided back to patients, it is clear prosthetic components can be repaired and reused. As many prosthetic components are overdesigned, there is still significant (if, yet, unquantified) life left in at least some prosthetic components at the time of replacement.

Many organizations collect and donate lower-limb prosthetic components to LMICs. For example, Humanity & Inclusion's Liimba project (launched in 2006) collects and refurbishes used prosthetic components from Belgium, France, Luxembourg, and Switzerland – where prosthetic components cannot be reused within country – for donation in countries where Humanity & Inclusion operates (https://www.hi.org/en/liimba--giving-a-second-life-to-prostheses-thanks-to-refurbishment). Similarly, Limbs for Life Foundation collects and refurbishes used prostheses and prosthetic components from the United States – which again are usable but cannot be reused within country – and provides prostheses/prosthetic componentry to other countries through partnerships, including Human Engineers Inc. (Philippines), Thomas L. and Linda J. McCormack Foundation (Panama), Protesis Imbabura (Ecuador), 2ft Prosthetics (Dominican Republic, Mexico, Philippines, Tonga), and STAND (sub-Saharan African) (https://www.limbsforlife.org). Another US organization, Hope to Walk, provides its own prosthetic legs for LMIC contexts, but also accepts and donates recycled prosthetic legs and parts ("If you would like to donate old prosthetic legs and parts to be used for these patients, please contact us." https://www.hopetowalk.org).

STAND, formerly Legs4Africa, is a UK-based charity that has collected 14,000 + donated, used prosthetic legs from individuals and P&O centres across US, Canada, UK, and across continental Europe. For feet to be accepted by STAND, they must have no visible cracks, wear, or degradation and be complete with no missing parts. They also must be modular prostheses, 1) ensuring easy dis- and re-assembly, 2) allowing for multiple brands and types of technology to be combined, and 3) enabling greater flexibility in alignment/adjustment throughout the prosthetic fitting/repair process for the prosthetists in the network. Prostheses collected in the US, Canada, and UK are shipped to Bristol where they are disassembled into their components (sockets, pylons, feet, and/or knees), inspected, boxed, and relevant parts are shipped to partner clinics in western and eastern Africa. In eastern Africa, components are shipped to Dar es Salaam, Tanzania, and redistributed to rehabilitation centres in Kenya, Tanzania, and Uganda. Prostheses collected in continental Europe are shipped to Nav Solidaire in Normandy, France; Nav Solidaire is STAND's collection partner in France and performs similar activities to STAND in Bristol (visual inspections and cleaning of the components). However, their operation involves shipping the components southwest across the Atlantic to The Gambia and Senegal. A carbon impact assessment conducted by an independent assessor estimated STAND saves 5.7 + tonnes of $CO_2e$ emissions/year by rescuing compared to manufacturing new parts.

As there are no standards and regulatory requirements for used prosthetic componentry, STAND initially disassembled modular prostheses and quickly visually inspected components for damage. However, some components – mostly feet and foot shells – would break during or shortly after shipment. To improve quality, prosthetic feet underwent additional scrutiny from January 2024 onwards, including:

1] Checking feet were complete and bumpers were present,

2] Inspecting for surface flaws including cracks, degradation, or discolouration that may affect cosmesis or indicate the onset of foot deterioration,

   A] During surface inspection, toes were bent and thumbs were pressed into the foot shell to reveal any hairline cracks that were otherwise invisible.

   B] Toe flexibility was also noted when toes were bent to ensure they were not overly flexible.

3] Ensuring non-SACH feet had matching keels and shells (size, model, and brand)

If the feet failed any one of these criteria, the foot was deemed unsuitable for donation and discarded.

As a first step to understand what type of international guidelines, standards, or regulations may be required for the reuse of prosthetic components, we examined the efficacy of the organizational standards used by STAND, with the idea that proven efficacy of these standards could lay the groundwork towards the development of future international standards. Our underlying scientific hypothesis is that the implementation of these standards will lead to an increase in the proportion of usable prosthetic feet. As feet can be damaged during shipping, we only review prosthetic feet already at prosthetic and orthotic centres in Africa, and not those that have yet to be shipped in Bristol.

## Materials and methods

In February 2025, MB and PM travelled to Fort Portal, Uganda and reviewed prosthetic feet donated by STAND. 196 feet were stored at the Ninsiima Centre for the Rehabilitation of People with Physical Disabilities (NCRPPD) and 170 a few miles away at Knowledge for Change (K4C). All feet were stored in cupboards at room temperature. The exact length of storage is unknown, but NCRPPD feet were stored in Uganda for longer (1+years) and were not inspected using STAND's checklist prior to shipment while K4C feet were stored in Uganda for less time (<1 year old) and were inspected using the checklist prior to shipment. Anything about the feet prior to donation (user, length/location of use, etc.,), and the length of time between replacement of the prosthetic foot in the P&O clinic in the US/Europe, donation, and arrival of the foot in Uganda is unknown.

The outer surfaces of the feet stored at NCRPPD and K4C were visually inspected by PM (a trained prosthetist), and data were recorded/feet photographed by MB. Feet were visually inspected and manually manipulated (bent and indented with fingertips) to identify completeness of the foot, stripped bolts/screws, broken keels, cracks (e.g., in the keel, foot shell, or other componentry), surface degradation or compliance in the polymer components (i.e., hyperelasticity), excessive cosmetic surface flaws (see Results), and presence of bumpers and keel stability within the foot shell, if applicable. The presence of any of these flaws would indicate the foot was likely to fail immediately or soon after being provided to the user, indicating the foot was "unusable". The lack of such flaws indicated the foot was "usable".

Occasionally, keels were removed from foot shells to identify prosthetic foot brand, but this was avoided when possible as keel removal can damage foot shells; inner surfaces of foot shells were not inspected for this reason. Foot side (left/right), size, model (i.e., solid ankle cushioned heel – SACH – vs other), brand, and defects were recorded. SACH feet derive their mechanical stiffness from both the harder inner keel and softer flexible surrounding material and can be used in modular prostheses with the correct ankle attachment. "Other" feet were modular, often flexible keel prostheses, but included some axial, dynamic response (Energy Storing And Return; ESAR), and defunct hydraulic prosthetic feet. This category consisted of keels and generally removable foot shells, although some keels were integrated into foot shells in a way that prevented their removal. Brand model (e.g., Steeper's Kinterra 3.0) was not recorded. No mechanical tests were run to investigate structural safety.

Multivariable binomial logistic regressions were run using the glm function in R/RStudio [38,39] to predict whether feet were usable (i.e., could be provided to prosthetic users). An initial regression was run including site of prosthetic foot storage, whether the foot was a left/right, foot size, foot model, and brand. A series of reduced statistical models considering each parameter, independently, were then run, followed by a final model which considered on the parameters from model 1 that were statistically significant (p<0.05). The seven statistical models were compared using Akaike information criterion (AIC), and the best fit model was used for data interpretation.

Prosthetic foot storage was used to investigate the efficacy of the checklist – as the feet stored at NCRPPD had not been checked before leaving Bristol, but those at K4C had, if those at K4C were more usable, this would imply the checklist is efficient. Foot side, foot size, model, and brand were included to investigate if any of these parameters affected the ability to determine foot usability and therefore affected the checklists performance. Side and size were included as diagnostic variables; neither was expected to affect usability. Model was included as SACH feet are more common in LMICs, and different brands may be more susceptible to damage during shipping, passing the checklist in Bristol but failing during transport.

**PLOS Global Public Health**

The performance of the best statistical model was investigated by binning residuals, examining Cook's distance, conducting a Hosmer-Lemeshow goodness-of-fit test, and calculating the dispersion parameter and area under the curve (AUC). Power analyses were run to investigate the effect of brand, which was significant, given the high number of brands relative to the overall sample size.

## Results

A total of 366 feet were analysed, 196 left and 170 right feet ranging in size from 15 to 30 cm. 59/366 feet were SACH, and a brand could be identified for 307/366 feet. The 307 feet represented 12 identifiable brands, the majority (60.66%) being Blatchfords (n = 120) or Ottobock (n = 102; Table 1; raw data in S1 Data). No damage was observed on any keels, but the structural integrity of the keels could not be determined as they were not mechanically tested.

Binomial logistic regressions revealed site (NCRPPD vs. K4C) and brand were both important factors, and the statistical model that included these parameters. There is strong evidence that models including foot side, size, and model did not affect foot usability (dAIC > 4; Table 2). Statistical model m7 (weight = 0.915; Table 2), which only included parameters site and brand, was used for further analysis. Model diagnostics demonstrated that, when BioQuest was coded as "zBioQuest" making Blatchfords the first alphabetically, model m7 was well-calibrated, stable, undistorted by outliers, not overdispersed, and predictively useful (see S1 File for explanation).

The feet stored at K4C were more usable than those stored at NCRPPD (Table 3). When accounting for the effect of brand (model m7), it was 102.5% more likely a foot stored at K4C was usable compared to NCRPPD (odds ratio = 2.025).

The effects of brand on foot usability was examined further. The global effect of brand was quantified using a likelihood ratio test, and the power effect of brand was significant (p < 0.001, degrees of freedom = 12, deviance = 34.043). To examine the effect of certain brands, the coefficients from m7 were extracted (Table 4). Some brands were represented by few

**Table 1.  Brands of feet available for prosthetic users in Fort Portal, Uganda. Sample (number usable, percent usable).**

|  | BioQuest | Blatchfords | College Park Industries (CPI) | Fillauer | Janton | Kingsley Casual | Össur |
|---|---|---|---|---|---|---|---|
| NCRPPD | 0 (0, NA%) | 58 (56, 96.55%) | 16 (12, 75%) | 4 (4, 100%) | 1 (1, 100%) | 2 (2, 100%) | 14 (9, 64.29%) |
| K4C | 1 (1, 100%) | 62 (62, 100%) | 11 (11, 100%) | 5 (5, 100%) | 0 (0, NA%) | 4 (4, 100%) | 11 (9, 81.82%) |
| Total sample | 1 (1, 100%) | 120 (118, 98.33%) | 27 (23, 85.19%) | 9 (9, 100%) | 1 (1, 100%) | 6 (6, 100%) | 25 (18, 72%) |
|  | Ottobock | Steeper | Streifeneder | Trulife | WillowWood | Unknown |  |
| NCRPPD | 59 (45, 76.27%) | 4 (4, 100%) | 2 (1, 50%) | 12 (8, 66.67%) | 7 (6, 85.71%) | 17 (15, 88.24%) |  |
| K4C | 45 (43, 95.56%) | 1 (1, 100%) | 0 (0, NA%) | 10 (8, 80%) | 1 (0, 0%) | 19 (16, 84.21%) |  |
| Total sample | 104 (88, 84.62%) | 5 (5, 100%) | 2 (1, 50%) | 22 (16, 72.73%) | 8 (6, 75%) | 36 (31, 86.11%) |  |

**Table 2.  Statistical models fit to the gathered data. + indicates the parameter was included in the model, while --- indicates the parameter was excluded. AIC values, difference in AIC (dAIC) values, and Weights were used to select the best fit model.**

| Model | Site | Side | Size | Model | Brand | AIC | dAIC | Weights |
|---|---|---|---|---|---|---|---|---|
| m7 | + | --- | --- | --- | + | 247.71 | 0 | 0.915 |
| m1 | + | + | + | + | + | 253.582 | 5.872 | 0.049 |
| m5 | --- | --- | --- | --- | + | 254.506 | 6.796 | 0.031 |
| m2 | + | --- | --- | --- | --- | 257.753 | 10.043 | 0.006 |
| m6 | --- | --- | --- | + | --- | 268.686 | 20.976 | 0 |
| m4 | --- | --- | + | --- | --- | 268.802 | 21.092 | 0 |
| m3 | --- | + | --- | --- | --- | 268.789 | 21.079 | 0 |

**Table 3. Usable and broken feet per site.**

| Place | | Count | Percent |
|---|---|---|---|
| The Ninsiima Centre for the Rehabilitation of People with Physical Disabilities | Broken | 33 | 16.84 |
| | Usable | 163 | 83.16 |
| Knowledge for Change | Broken | 10 | 5.88 |
| | Usable | 160 | 94.12 |

**Table 4. Effect of brand on prosthetic foot usability. Coefficients (which are log-odds differences for brands relative to Blatchfords) are taken from model m7 and measured relative to brand "Blatchfords" (see S1 File and S1 Data). Higher values indicate a larger proportion of the feet were usable.**

| | Coefficient | Sample (percent usable) |
|---|---|---|
| Janton | 13.917 | 1 (100%) |
| Steeper | 13.75 | 5 (100%) |
| Fillauer | 13.407 | 9 (100%) |
| Kingsley Casual | 13.282 | 6 (100%) |
| BioQuest | 12.811 | 1 (100%) |
| Blatchfords | 0 | 120 (98.33%) |
| College Park Industries (CPI) | -2.25 | 27 (85.19%) |
| Unknown | -2.299 | 36 (86.11%) |
| Ottobock | -2.322 | 104 (84.62%) |
| WillowWood | -2.655 | 8 (75%) |
| Trulife | -3.101 | 22 (72.73%) |
| Össur | -3.123 | 25 (72%) |
| Streifeneder | -3.649 | 2 (50%) |

feet (i.e., < 20) and the results for these were ignored. It should further be noted that some P&O centres preferentially use certain brands, and the P&O centres from which the donations originated was not recorded. As such, the statistical effect of "brand" may represent something related to the brand of the foot (e.g., material, model, manufacture methodology) or something related to the initial provision of the foot.

Blatchfords feet performed the best, with 98.33% of the 120 feet being usable (Table 4). This was followed by College Park Industries (CPI, 85.19%), Ottobock (84.62%), Trulife (72.73%), and Össur (72%), but the samples were small (<30 feet) for most brands. As m7 fit the data much better than m1, this implies foot side (left/right) and size did not affect usability. Similarly, SACH vs. "other" models did not affect usability at the time of donation.

Causes of foot failure (i.e., foot unusability) included the foot shell having cracks, and being crumbly, discoloured, worn, or sticky due to polymer degradation (see Fig 1), with cracks being the most common cause of failure (Fig 2a). Additionally, the keel could have been loose, or the foot could have been exceedingly stretchy. Of the 43 broken/unusable feet, three had two mechanisms of failure and the remaining 40 had one. Samples were not large enough to investigate the interaction between type of failure and prosthetic foot brand.

When causes of foot failure were divided by centre, the feet at NCRPPD (Fig 2b) experienced all forms of failure except wear while the feet at K4C (Fig 2c), which had gone through a usability inspection in Bristol, only failed due to cracks, crumbling, or wear. PM, who carried out the inspection for STAND in Bristol, stated the one foot with wear should have been removed using the inspection system in place, and its shipment to Uganda was user error.

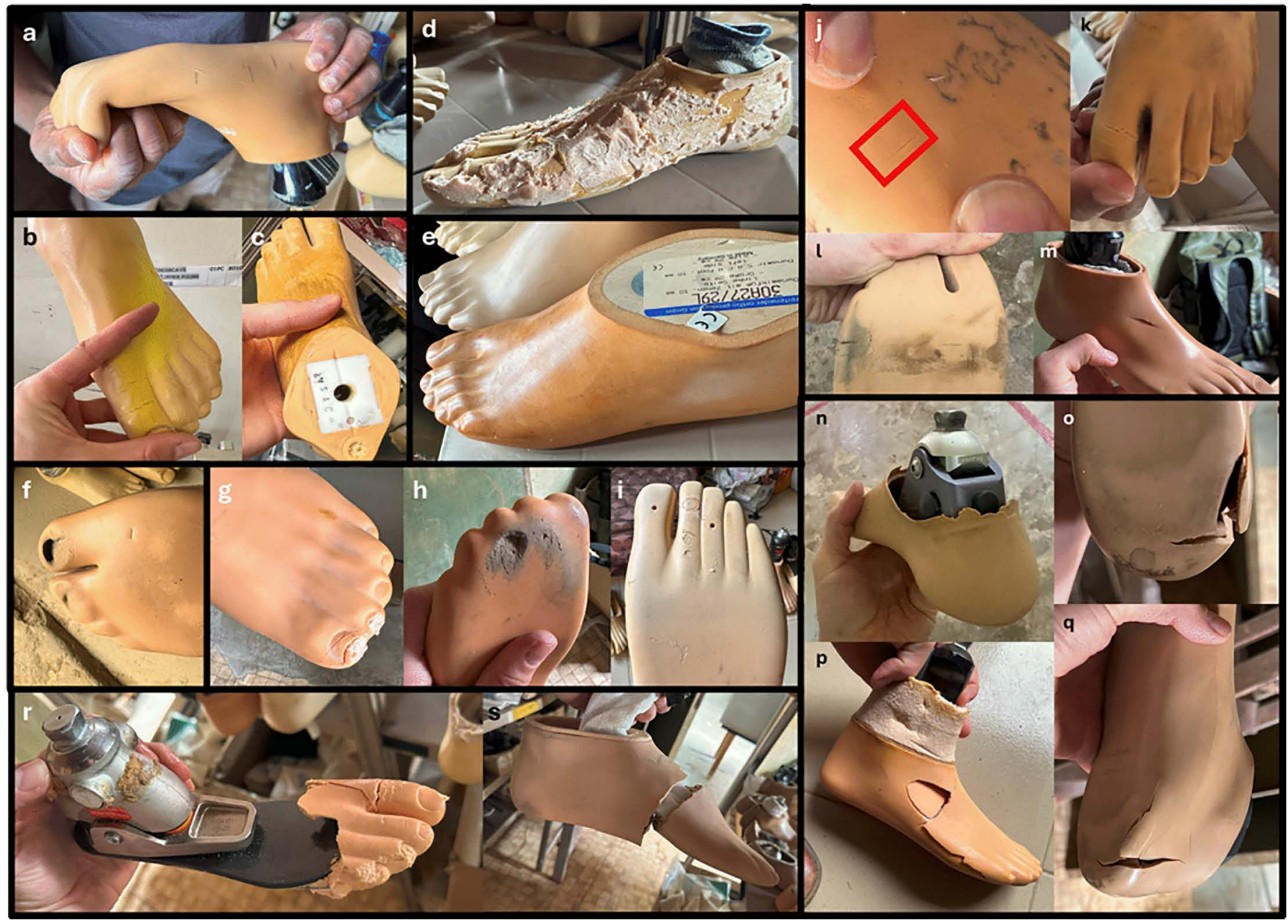

**Fig 1. Examples of how feet could become unusable.** a) polymer degradation, extreme flexibility in toes, b) discolouration and cracks on first toe, c) discolouration, d) a structurally stable (and therefore, usable) foot with cosmetic faults due to being covered in foam, e) degraded surface which became sticky to touch, f) foot shell worn through under toes, g-i) foot shells crumbling, j-m) small cracks, n-q) large cracks/fractures, r-s) catastrophic cracks/fractures causing mechanical failure.

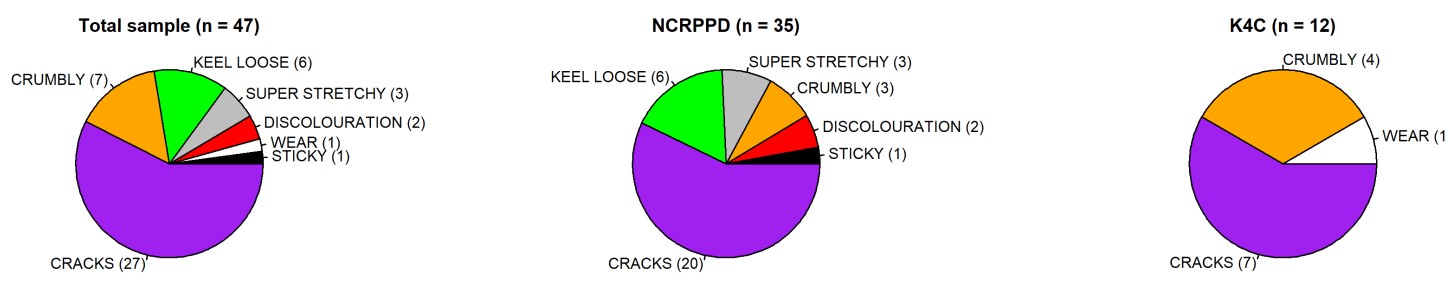

**Fig 2. Issues which made prosthetic feet unusable, total and divided by centre.**

Finally, differences in foot failure were examined between SACH and other feet, as SACH feet are commonly manufactured and provisioned in LMICs (e.g., by the International Committee of the Red Cross or Exceed Worldwide). SACH feet tended to become cracked, crumbly, or sticky, while other feet experienced all forms of foot failure (Fig 3).

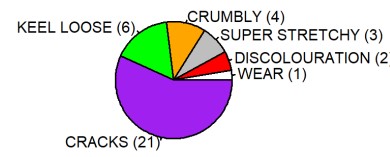

**SACH feet (n = 10)**

CRUMBLY (3)

STICKY (1)

CRACKS (6)

**Other feet (n = 37)**

KEEL LOOSE (6)

CRUMBLY (4)

SUPER STRETCHY (3)

DISCOLOURATION (2)

WEAR (1)

CRACKS (21)

**Fig 3. Issues with SACH vs other prosthetic feet.**

## Discussion

Donated assistive technologies can be a major benefit to LMICs provided they meet standards and regulatory requirements. No (inter)national standards or regulatory requirements exist for used prosthetic devices, particularly those that are out of warranty. This can lead to substandard devices being supplied to LMICs, which can be a burden to the recipient LMIC [32,40,41].

> "[The] 'Dumping' of obsolete equipment by high-income countries (HICs) has been described as 'morally reprehensible' and has received adverse media attention. An 'anything is better than nothing' attitude, coupled with a donor–recipient power imbalance, has been cited the central reason for poor-quality donations. Recipients may be too embarrassed to point out the futility of donor efforts or may find it culturally inappropriate to decline a gift. As a result, 'medical equipment graveyards' of obsolete or broken donated biomedical equipment are commonly seen in hospitals across low-income and middle-income countries (LMICs) [32]."

To prevent morally reprehensible actions and the creation of prosthetic landfills, standards, like those in WHO's principles and guidelines for the donation of medical equipment, can be applied to prosthetic devices [28,29,31]. The guidelines for donation clearly set out donor and recipient responsibilities for the donation of medical devices (Fig 2 in section 3.3; World Health Organization, 2024). The responsibilities of, e.g., understanding national regulatory requirements, conducting functionality tests, and submitting proof of functionality, lie with the donor. Conversely, the responsibility to refuse incomplete medical devices and ensure proper management, storage, and use of those devices lies with the recipient. The Medicines & Healthcare products Regulatory Agency (MHRA) in the UK considers external prosthetic devices to be constructed of 2 parts, the socket and the "hardware" (i.e., all components up to the socket). Should recipients refuse donations of prosthetic components if they do not have access to the material (e.g., polypropylene) to manufacture sockets? Should donations of used prosthetic components be accompanied by access to supply chains for material to manufacture sockets?

Donations should take place within the framework of health partnerships where mutual responsibilities are clearly established and understood. This includes discussions about resources, supply chains, and skills required to manufacture and maintain prostheses. Receiving organisations may not be able to access public funding to, e.g., locally manufacture socket, and may expect the donor to fund this. These organisations must understand and respect the context in which items are received and that donors have finite funding and options. If funds are not provided, receiving organisations may plan to charge patients to cover associated costs. Partners need to negotiate carefully to ensure that their expectations are fully understood, and accountability systems need to be implemented to evaluate prosthetic provision. This should include clear statements about the sale/resale of donated componentry and whether cost-sharing with end-users is permitted. Ethical considerations extend past WHO's principles, and must consider equitable, dignified, and inclusive prosthetic provision where, like in the WHO's principles, the responsibilities of the donors and recipient are clearly stated. The equitable provision of prosthetic devices must be considered during device collection, checks, shipment, and dissemination (to the P&O centres and prosthetic user). At the point of collection, questions such as could the device be better used

by the community it is being removed from? And could better or affordable versions of the same technologies be supplied new or manufactured locally in the recipient country?

All devices should be inspected, checked, and shipped to the same standard, regardless of donor or recipient. The WHO guidelines state that the donor and recipient should discuss quality and quantity and that the donated equipment should be functional for 2+years post-donation for sustainable donation; however, a preponderance of lack of compliance with these guidelines has been found [42]. Strict compliance with the official WHO guidelines has been recommended for sustainable donation programmes involving medical devices and equipment [25,40,42,43].

Choice of country to ship devices to is important, not only considering the prosthetic user needs, but also the capabilities of the donors and recipient organisations in those settings. Having an established network in a country with relatively lower need may trump providing devices to a country with a higher need but a worse network, as the former may ensure devices are supplied to users that need devices but the latter may not. Devices should be supplied to P&O centres in an equitable manner that reflects not only user needs, but the operation of the centre itself and to the patients on the basis of need and not necessarily ability to pay or simply first-come first-served [5].

Mutual respect between donors and receiving organisations and the dignity of the end users is paramount. As discussed previously, receiving organisations must be actively engaged in discussions about proposed donations and should not feel under pressure to accept donations, although it should be recognised that recipients may feel they must accept all donations in a grateful manner to ensure continued support [32]. This can be undignified for the recipients, who may feel the donations are not useful. Similarly, the devices themselves may have social or cultural issues (e.g., related to cosmesis) and prosthetic users may feel it is undignified to use these prosthetic devices, potentially leading to abandonment [11,13,44]. Although abandonment rates in higher income settings for some types of prosthesis and other assistive technologies are well-established [45–47], this issue has received little attention in other settings. Longer term, if donor organisation were required by their funders/supporters to report not only the number of prostheses donated, but also the abandonment rates, this might focus attention on appropriateness of donations as well as the quantity. Social, cultural, etc., factors [4] must be considered when ensuring dignified provision. As these devices are basic human rights, users should not be made to feel they are fortunate or lucky to receive the devices.

Obligations of the donors should be considered across time scales. On the short scale, donors should at a minimum ensure they are providing technologies that work, can be used by the recipient (correct shape, size, and provisioned by prosthetists/technicians who are trained in that technology), are fulfilling a need, and are wanted by the recipient. On the long scale, donors should ensure the systems are robust and work well [14,30,32]. When creating a new supply chain (i.e., donations), it is the ethical responsibility of the donors to understand what supply chains they are replacing and how they will ensure supply chains continue when the donations stop, ensuring the system is sustainable and does not prevent local innovation [5,6]. Given the manufacturer of devices will likely only provide user manuals in languages relevant to their key export markets, careful consideration should be given to how such information can be made accessible to recipients in other countries [35].

Clear standards exist for prosthetic design (e.g., ISO 10328 and 22675), manufacturing (e.g., ISO 9001), and dissemination (e.g., WHO prosthetics should be available for all). "Poor quality assistive products exist due to inadequate standards, lack of regulatory enforcement and lack of knowledge about the need for safe and effective products [21]." What standards/regulatory requirements should exist for prosthetic donation/provision?

If donated prosthetics components are new and not yet used, they should meet the standards for prosthetic provision of the donor country. But if used, separate standards need to be applied.

## Towards a set of standards for the donation of used prosthetic feet

Here, we investigated an internal checklist used by STAND to work towards a set of standards to govern the donation of prosthetic feet. While they are consistent with many of the WHO's indicators of suitability for medical device donation (i.e.,

appropriate for setting, assured quality and safety, affordable and cost-effective; [28]), it should be stressed this is only a first step towards the creation of a set of standards for the reuse of prosthetic feet, as it does not include factors such as mechanical/structural testing, end-user trials, or patient-level outcomes. This method must also be rigorously tested for sensitivity and specificity and, if necessary, appropriately altered. [28,29,31]

Currently, we suggest these standards be taken on as internal operating procedures by donors, and stress that the adherence to these standards is the responsibility of the donors and not the recipients. This is also consistent with many attributes of the WHO's principles of good donation, which puts the responsibility on the donor to carry out these checks (i.e., conducting functionality tests) before the prosthetic feet are donated to the recipient, and require some proof of functionality to be submitted with each prosthetic foot [28,29,31]. In the future, we recommend any future alterations, additions, or substitutions to these standards should also incorporate structural testing and potentially simplified versions of quality assurance tests carried out by manufacturers. We further recommend the creation of formal, international regulatory requirements/standards to govern prosthetic component donation be created, to ensure quality and prosthetic component safety.

STAND's internal inspection protocol increased the quality of prosthetic foot provision and led to a 65.1% reduction in unusable feet donated to Uganda. However, unusable prosthetic feet that were cracking/crumbling were still being donated (Fig 2c), possibly because feet were missed during inspection or were experiencing further degradation after leaving Bristol. The inclusion of a chemical or mechanical polymer testing for prosthetic foot shells may be beneficial. An altered form of STAND's inspection method, informed our results, is presented in Table 5.

On the checklist, factors related to mechanical function should be treated as hard cut-offs; prosthetic components that cannot guarantee the user's physical health and/or will break soon after donation should not be provisioned. Factors related to cosmesis (e.g., Fig 1d) require judgement from the provisioner, and recipients should be allowed to turn down donations for cosmetic reasons [32] as they could cause psychological, social, and/or cultural issues. Of note, cosmetic factors – some of which are related to prosthetic component acceptance and abandonment, such as prosthetic component colour – are largely missing from Table 5. Similarly, economic, social, cultural, etc., factors [4] are not considered but should be included in prosthetic component donation standards/regulations. Investigations into quality assurance checks by prosthetic component manufacturers may help improve quality standards for used prosthetic components. The co-development of checklists with both donors and recipients may help improve donations [40].

**Repair, reuse, recycle, and systems surrounding the donation of used prosthetic feet**

Prosthetics can last long in LMICs [48–50] as they are overdesigned for their intended length of use, and prosthetics in HICs are overdesigned for their intended length of use, making them prime for repair, reuse, and recycle. In the UK, prosthetics and their components (e.g., prosthetic feet, titanium adapters) are often replaced due warranties or guidelines and not prosthetic component breakages (personal communication). Implementing a repair and reuse model is critical for prosthetic component donation and can help promote prosthetic component longevity in LMICs, particularly in rural areas, which have relatively more prosthetic users, lack P&O centres, and are difficult to reach [2,3,9,19]. Different barriers exist in implementing a repair, reuse, recycle model for prosthetic provision within and between LMICs. For example, in India, cost was the biggest barrier for upper limb body-powered prosthetic repair [44]. In Uganda, the biggest barrier to repair differs depending on who is consulted; from the prosthetist's perspective, it is materials, but from the patient's perspective, it is cost [5].

An integration of global and local markets, where prosthetic components are supplied internationally and repaired locally, may be an efficient way forward [6,21]. Global markets can drive down the cost of prosthetic components due to bulk manufacturing – assuming prosthetic components can be purchased at a level to drive down the cost [21] – but are likely to be disrupted by global crises, like the Covid-19 pandemic [6]. The combination of many different technologies, requiring large levels of stock for spare parts and large skillsets ranging all possible products, however, may be

**Table 5. Suggested checklist to ensure quality and standards for donation of prosthetic feet.**

| Instructions: |  |
|---|---|
| 1) Visually inspect the foot using the checklist below. | |
| 2) Bend the toes forward and backward, checking for small cracks (Fig 1i-) at the points where the foot is being bent. | |
| 3) Press the foot at various locations all over the foot, looking for hairline cracks that may appear during localized deformation. | |
| **A "no" to checklist questions 1, 5, or 6, or "yes" to questions 2–4 indicates the foot has failed inspection and should not be donated.** | |
| **Checklist questions** | **Logic/reasoning** |
| 1) Is the foot complete? (Fig 1r) <br> Does the foot have all its components? For example, is the foot shell missing? | Incomplete feet are not usable. While parts can be used to fix broken feet, it is better practice to supply complete feet rather than parts for feet the P&O centre may not have |
| 2) Are there any flaws related to mechanical function? <br> Examples include stripped screws, broken/delaminated keels, cracks of any size and surface degradation of the polymer components (e.g., stickiness, crumbling; Fig 1e, Fig 1f-s) | Mechanical failures will prevent the prosthetic foot from being usable. Cracks or surface degradation, like stickiness or crumbling, can indicate polymer degradation and that the foot has mechanically failed or will begin to mechanically fail soon. This is particularly true for feet with removable foot shells. |
| 3) Is the foot compliant? (Fig 1a) | If too pliable, the foot will not be rigid enough to support a person's weight during locomotion and is prone to break |
| 4) Are there any excessive surface flaws related to cosmesis? <br> Examples include surface degradation (e.g., crazing; Fig 1c), discolouration (Fig 1b), inclusion of other materials (Fig 1d), and staining. | Cosmetic issues can lead to prosthetic abandonment. Some small flaws, like minor scratching or discolouring, can be considered acceptable. |
| For feet with removable foot shells | |
| 5) If the foot has bumpers, are they present? | Bumpers change foot stiffness, improve alignment, and enable smooth gait |
| 6) Are keels and foot shells correctly matched? | Incorrect matches between keels and foot shells can cause poor fits, leading the shell to move relative to the keel and/or detach during use |

problematic for local prosthetic repair. Staff shortages, large patient numbers, lack of components, and practical and financial challenges mean prosthetists in LMICs need to be more creative in technology creation and provision and likely require more training and skillsets than prosthetists in HICs [4]. Modular prostheses make the integration of different technologies easier, and reduce the necessary stock, but limit product variability, particularly with lower-limb prosthetic cosmesis. The ability to manually construct and repair prosthetics is important to successful prosthetic provision under the repair, reuse, and recycle model [2,6].

The integration of digital manufacturing and 3D printing into prosthetic provision and repair could aid in the local manufacture of prosthetic components, reducing the volume of spare parts that must be kept in stock and allowing the manufacture of parts for the repair of older devices that are no longer being manufactured. The digital manufacture of parts requires the establishment of local manufacturing centres, upskilling workers, creation of supply chains for both manufacture of prosthetic components and maintenance of equipment, 3D digital models of the parts to be manufactured – which organisations may not be willing to share – as well as the construction of adequate manufacturing facilities (i.e., rooms with consistent, reliable power operating within tolerable temperatures) [51].

A systematic review on the application of 3D printing to prosthetics in developing countries was conducted in 2021 [52]. They identified 14 studies from 2014-2020 that included a total of 47 prosthetic users, where devices were not always manufactured or tested in the developing country. The biggest advantages to 3D printing prosthetics were the ability to manufacture low-cost prostheses without loss of functionality and the ability to fit prosthetics at a site other than the place of manufacture. Prosthetics were generally cheap, but not always so. The biggest drawbacks were time to manufacture and that most studies did not conduct trials in LMICs. Not discussed were issues related to local infrastructure [32], supply chains and related trade barriers [53], maintenance and repair of equipment, etc. as "…unreliable supply of quality prosthetic devices and components is a significant hindrance, slowing or stopping fitting [12]". It was also unclear in the papers

reviewed what parts were 3D printed, what the lifespan of the devices were, etc., [52]. It appears that, as far as 3D printing prosthetics in LMICs is concerned, device design is the focus, and most other related factors are ignored [54].

Reusing prosthetic components can lead to a reduction of waste, but inappropriateness or user dissatisfaction in assistive technologies can lead to abandonment, increasing waste [6]. In Norway, 1/3 of assistive devices are reused through the Norwegian Assistive Technology Provision Model [6]. In other HICs re-use rates are highly likely to be lower; for example, the UK NHS has recognised that many walking aids do not get re-used [55]. However, technologies are not always appropriate to reuse in LMICs, and the "something is better than nothing" paradigm is not always true [56]. Time should be given to study and understand the local environment and culture [4] – devices should be low cost, locally available, capable of manual fabrication, considerate of local climate and working, durable, simple to repair, simple to process using local production, reproducible by local personnel, technically functional and not too high-tech, biomechanically appropriate, lightweight, and cosmetic, psychosocially appropriate [4].

## Future work

The use of donated, used prosthetic limbs and their components has the potential to be a major source of prosthetic component supply. Donors should adopt minimum standards to ensure device quality and function, devices fulfil end-user needs (e.g., biomechanical, social, cultural), and devices work within local systems without displacing existing services. International standards to ensure quality should be created, potentially with an associated accreditation process operating within a regulatory environment.

To begin this journey, work towards a proposed set of standards in the form of a quality assurance check that focus on prosthetic foot quality at the point of provision (Table 5). However, the checklist proposed here has not yet been validated, and its sensitivity/specificity remains to be checked. Rapid, affordable, and non-/minimally destructive tests should be created to test the structural safety and/or lifespan of the prosthetic feet to promote prosthetic user safety, and patient-outcomes need to be considered. Structural test(s) that focus on prosthetic foot/keel integrity could take inspiration from ISO 10328 and/or 22675 tests. Given the issues observed with the foot shells (Fig 2a), non-/minimally destructive methods for quantifying polymer quality would be beneficial. Some critical data that are needed for the full development of standards/tests include:

• When/why are feet being replaced at the time of collection for reuse and recycle?

• How long do feet last once repaired and provisioned?

• What factors affect used prosthetic foot safety and lifespan?

• What differences exist in user quality of life between new and used prosthetic feet?

It is possible different standards will be required for different foot models because of differences in failure mechanisms. For example, SACH feet, where the keel is fully integrated into the prosthetic foot, did not suffer from keel loosening, but other prosthetic foot models did (Fig 3).

Finally, the efficacy of the checklist was supported by a regression model which has not been validated. This should be done, and factors related to usability, such as what was designated "brand" in this study, should be investigated further to identify if usability is, for example, related material, prosthetic component use, P&O centre policies, or length/magnitude of use prior to donation.

## Limitations

A lack of consistency in brands, and knowledge of models or where the prosthetic feet originated from, means it is not possible to make conclusions about which brands are better/worse for donation. While 366 feet is a large sample, it is also not large enough to determine, e.g., the effect of brand on usability, as the sample sizes for many brands are too small.

The feet at the NCRPPD have been stored at ambient temperature in Fort Portal for longer than those at K4C. Polymers, like the foot shells, degrade under warmer temperatures, which is why it is recommended prosthetic feet be stored at certain temperatures. However, the range of storage temperatures for prosthetic feet can be vast (e.g., -15 to 50°C for Blatchfords ELAN feet, models ELAN22L1S—ELAN30R8S and ELAN22L1SD—ELAN30R8SD) and, while temperature was not recorded in the storage facilities at these centres, it is unlikely to have exceeded 50°C. It is therefore possible some of the differences we observed in foot failure were due to differences in storage time in warm temperatures.

## Conclusions

Here, we work towards a proposed set of standards for the donation of used prosthetic feet. We have demonstrated the efficacy of our checklist, and hope it will lead to improved and consistent quality in donated prosthetic feet, particularly for LMICs. We stress that the responsibility of adhering to these standards lies with the donor and not the recipient. These standards do not consider many aspects important for prosthetic feet (e.g., structural stability) which should be considered in future additions/alterations to this methodology. The creation of standards/regulatory requirements governing the donation and provision of used prosthetic components is critical not only for the donation of prosthetic components to LMICs, but the creation of a circular economy surrounding prosthetic provision in HICs.

## Supporting information

**S1 Data. raw data used in this study.**
(CSV)

**S1 File. Electronic Supplementary Material (ESM) model diagnostics for the statistical model used in the results and discussion sections.**
(DOCX)

## Acknowledgments

We would like to thank STAND, Knowledge for Change, The Ninsiima Centre for the Rehabilitation of People with Physical Disabilities, British Academy for working with us on this project and/or funding this research. We would like to thank members of STAND for information on the carbon footprint of their work and for comments on data collection/analysis. We would further like to thank Prof Nachiappan Chockalingam for comments on an early version of this manuscript. Finally, we would like to thank the editor, two anonymous reviewers, and Dr Margrit R Meier for their comments and help in improving this manuscript.

## Author contributions

**Conceptualization:** Michael A Berthaume, Laurence Kenney, Vikranth Harthikote Nagaraja.

**Data curation:** Michael A Berthaume.

**Formal analysis:** Michael A Berthaume.

**Funding acquisition:** Louise Ackers.

**Investigation:** Michael A Berthaume, Louise Ackers, Laurence Kenney, Promise Maduako.

**Methodology:** Michael A Berthaume, Promise Maduako.

**Project administration:** Michael A Berthaume.

**Resources:** Louise Ackers, Laurence Kenney, Promise Maduako.

**Visualization:** Michael A Berthaume.

**Writing – original draft:** Michael A Berthaume, Louise Ackers, Laurence Kenney, Vikranth Harthikote Nagaraja, Promise Maduako.

**Writing – review & editing:** Michael A Berthaume, Louise Ackers, Laurence Kenney, Vikranth Harthikote Nagaraja, Promise Maduako.

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
