## [Decision Letter · Decision Letter 0]

30 Dec 2025

PGPH-D-25-02840

Proposed standards for prosthetic foot reuse and considerations for donation of used prosthetic feet to low-and middle-income countries

Dear Dr. Berthaume,

Thank you for submitting your manuscript to PLOS Global Public Health. After careful consideration, we feel that it has merit but does not fully meet PLOS Global Public Health’s publication criteria as it currently stands. Therefore, we invite you to submit a revised version of the manuscript that addresses the points raised during the review process.

The manuscript has been evaluated by three reviewers, and their comments are available below.

The reviewers have raised a number of concerns that need attention. In particular, they request additional information on the development and validation of the checklist, and attention to the statistical analysis and interpretation of the results.

Could you please revise the manuscript to carefully address the concerns raised?

We look forward to receiving your revised manuscript.

Kind regards,

Helen Howard

Staff Editor

Journal Requirements:

1. Please provide a detailed online Financial Disclosure statement. This is published with the article. It must therefore be completed in full sentences and contain the exact wording you wish to be published.

a) Please clarify all sources of financial support for your study. List the grants, grant numbers, and organizations that funded your study, including funding received from your institution. Please note that suppliers of material support, including research materials, should be recognized in the Acknowledgements section rather than in the Financial Disclosure.

b) State the initials, alongside each funding source, of each author to receive each grant. For example: “This work was supported by the National Institutes of Health (####### to AM; ###### to CJ) and the National Science Foundation (###### to AM).”

c) State what role the funders took in the study. If the funders had no role in your study, please state: “The funders had no role in study design, data collection and analysis, decision to publish, or preparation of the manuscript.”

For more information, please go to our submission guidelines:

https://journals.plos.org/globalpublichealth/s/submission-guidelines#loc-financial-disclosure-statement

2. Please ensure that the funders and grant numbers match between the Financial Disclosure field and the Funding Information tab in your submission form. Note that the funders must be provided in the same order in both places as well.

3. Please send a completed ‘Competing Interests’ statement, including any COIs declared by your co-authors. Please declare all competing interests beginning with the statement “I have read the journal's policy and the authors of this manuscript have the following competing interests:”

For more information, please go to our submission guidelines:

https://journals.plos.org/globalpublichealth/s/submission-guidelines#loc-competing-interests

4. We note that your Data Availability Statement is currently as follows: “All data is available in the ESM”

Please confirm at this time whether or not your submission contains all raw data required to replicate the results of your study. Authors must share the “minimal data set” for their submission. PLOS defines the minimal data set to consist of the data required to replicate all study findings reported in the article, as well as related metadata and methods (https://journals.plos.org/globalpublichealth/s/data-availability#loc-minimal-data-set-definition).

If your submission does not contain these data, please either upload them as Supporting Information files or deposit them to a stable, public repository and provide us with the relevant URLs, DOIs, or accession numbers. For a list of recommended repositories, please see https://journals.plos.org/globalpublichealth/s/recommended-repositories.

5. Please provide separate main figure files in .tif or .eps format only and ensure that all files are under our size limit of 10MB.

6. We have noticed that you have uploaded Supporting Information files, but you have not included a list of legends. Please add a full list of legends for your Supporting Information files before or after the references list.

7. Some material included in your submission may be copyrighted. According to PLOS’s copyright policy, authors who use figures or other material (e.g., graphics, clipart, maps) from another author or copyright holder must demonstrate or obtain permission to publish this material under the Creative Commons Attribution 4.0 International (CC BY 4.0) License used by PLOS journals. Please closely review the details of PLOS’s copyright requirements here: PLOS Licenses and Copyright. If you need to request permissions from a copyright holder, you may use PLOS's Copyright Content Permission form.

Potential Copyright Issues:

Figure 1: Please confirm (a) that you are the photographer; or (b) provide written permission from the photographer to publish the photo(s) under our CC-BY 4.0 license.

Additional Editor Comments (if provided):

Reviewers' comments:

Reviewer's Responses to Questions

**Comments to the Author**

1. Does this manuscript meet PLOS Global Public Health’s publication criteria? Is the manuscript technically sound, and do the data support the conclusions? The manuscript must describe methodologically and ethically rigorous research with conclusions that are appropriately drawn based on the data presented.? Is the manuscript technically sound, and do the data support the conclusions? The manuscript must describe methodologically and ethically rigorous research with conclusions that are appropriately drawn based on the data presented.

Reviewer #1: No

Reviewer #2: Yes

Reviewer #3: Partly

2. Has the statistical analysis been performed appropriately and rigorously?

Reviewer #1: No

Reviewer #2: Yes

Reviewer #3: Yes

3. Have the authors made all data underlying the findings in their manuscript fully available (please refer to the Data Availability Statement at the start of the manuscript PDF file)?

The PLOS Data policy requires authors to make all data underlying the findings described in their manuscript fully available without restriction, with rare exception. The data should be provided as part of the manuscript or its supporting information, or deposited to a public repository. For example, in addition to summary statistics, the data points behind means, medians and variance measures should be available. If there are restrictions on publicly sharing data—e.g. participant privacy or use of data from a third party—those must be specified.requires authors to make all data underlying the findings described in their manuscript fully available without restriction, with rare exception. The data should be provided as part of the manuscript or its supporting information, or deposited to a public repository. For example, in addition to summary statistics, the data points behind means, medians and variance measures should be available. If there are restrictions on publicly sharing data—e.g. participant privacy or use of data from a third party—those must be specified.

Reviewer #1: Yes

Reviewer #2: Yes

Reviewer #3: Yes

4. Is the manuscript presented in an intelligible fashion and written in standard English?

Reviewer #1: No

Reviewer #2: Yes

Reviewer #3: Yes

Reviewer #1: 1. The manuscript primarily shows that adding a visual inspection step increased the proportion of prosthetic feet deemed usable (83% to 94%). This outcome is predictable and does not constitute meaningful scientific innovation. The work reads as an operational description rather than rigorous research; novelty and contribution are therefore limited.

2. The proposed checklist is not validated. There is no mechanical or structural testing, no clinical functional outcomes, no prospective field evaluation, no inter-rater reliability assessment, and no sensitivity or specificity analysis. Accordingly, the checklist cannot be considered a standard, and the conclusions overstate the evidence. A formal validation phase is required.

3. Safety, mechanical integrity, and lifespan have not been evaluated. Visual inspection alone is inadequate for medical devices. No ISO-aligned static or cyclic loading tests are presented, nor are durability or time-in-service data available. This is a critical omission given the manuscript’s intent to inform international practice.

4. No patient-level outcomes are included (for example, fit success, comfort, skin issues, mobility, abandonment, repair frequency, or time-to-failure). Without these data, the practical value of the intervention remains uncertain.

5. Brand-level comparisons are underpowered, and model-level or material-level analyses are not presented. Despite acknowledging this limitation, the manuscript still interprets brand-related effects.

6. The Introduction and narrative sections are disproportionately long and repetitive; substantial condensation is recommended. In contrast, the Methods and Results require greater depth and clarity.

7. The statistical analysis is limited. Logistic models do not account for key confounders such as service age, storage duration, materials, or model type. Model diagnostics, effect sizes with confidence intervals, and multiple-comparison considerations are not reported.

8. Economic evaluation is absent. Donation and reuse programs in low and middle income settings are cost sensitive, and without cost modeling, the recommendations have limited actionable value.

9. Several claims are overstated, including suggestions related to circular economy effects, international standard development, and safety assurance. These assertions are not supported by the presented data and should be moderated.

Reviewer #2: It is suggested to review the Nippon Foundation/Exceed Cambodia in proposing the standards of P&O. The case study that has been done in Cambodia, Myanmar, Laos, Vietnam and Sri Lanka in will guide the current P&O Standard in low and middle income countries.

It is best to review the minimum standards of P&O in these countries as a underlying theory to govern the foundation of foot reuse and donation used.

A robust systematic reviews are vital in proposing standards for foot reuse and donations used in low and middle income countries. An updated literature are needed.

It is suggested to explore the preliminary findings in these low and middle income countries.

Reviewer #3: GENERAL

This reviewer welcomes the ambition of the authors to start developing standards for donated prosthetic componentry to LMICs. Such standards are indeed much needed as one important factor to improve the quality of the prosthetic devices provided within LMICs.

The authors’ work has carefully been imbedded into a wealth of information and reasons for why the need is urgent for developing standards of donated prosthetic components. This information has been mindfully drafted including viewpoints and situation of many LMICs as well as HICs. Well done!

What left this reviewer wondering is why the development of the checklist has not been carried out with locals at the two centers, where MB and PM were able to collect the data of the stored feet. The rationale for not doing so should be included into the Limitations section.

Further, why has no testing of the developed checklist been carried out with the two centers? For example, dividing the available feet into two equal sized groups would have raised the opportunity to develop the checklist with one group of feet including the regression model and then test it on the remaining feet in the second group. Why was this not considered? One could classify all available feet as indicated in Table 1, but then consider only these feet who were mostly used in the field or were mostly available. Lowering the numbers of independent variables to the those variables that would represent the essence of the checklist best would have given the option for a regression model, or is this reviewer mistaken? These points should be discussed in the paper. In case the paper gets too long (word count), it is recommended to concise the actual discussion section as it provides similar points stated in the introduction.

And lastly, this reviewer does not think that retesting used feet similar to the stated ISO standards would be feasible. Instead, it might be worthwhile checking in other industries (aviation, deep-sea shipping) what type of non-mechanical controls for checking of wear and tear on materials/motors are available without dismantling motors or testing of used structures. Perhaps some light and/or sonar evaluation would be a way to check the mechanical structure of used prosthetic feet and other componentry without putting any more strain on the used materials. That might be some thoughts for the Future Work section. Also probable collaboration with universities in LMICs should be considered as a close source of additional brain power for the development of standards within a given country.

DETAILED

The reviewer finds the word ‘prosthetics’ difficult and prefers the (correct) term ‘prosthetic componentry or prosthetic components’ instead. In her experience using the nomenclature of the P/O profession adds clarity in an interdisciplinary context. It is often unclear to people outside of or adjacent to the P/O profession that a ‘prosthetics’ is composed of different products, i.e. some industrial produced prosthetic components and – in most cases – a bespoken locally fabricated prosthetic socket. By using prosthetic components or prosthesis/prostheses when referring to the final product – the authors will signal directly that there are ‘pieces’ needed to compose an entire prosthesis. Further, using the correct term assists in distinguishing prostheses fabricated with componentry from those being fabricated by 3D printing, also a field needing standards for C2C design. Therefore, please change the wording accordingly within the entire paper – thank you!

Lines 165-168. This sentence seems to be incomplete – please check.

Line 229. This statement is incorrect. In Switzerland (and the reviewer is sure this is the case in France, Netherlands and the UK), prosthetic componentry has different life/warranty cycles depending on the type of prosthetic component and its model. Please rephrase this sentence pointing out that different prosthetic components and their models have different life/warranty cycles set by the industrial manufacturers.

Lines 284-286.This sentence is unclear: Are the authors checking prosthetic feet shipped to Africa prior to the study or as part of the study when these feet arrive in Africa? If they are analyzed prior to the study how do the authors make sure that the damage seen is indeed due to shipping and not due to storage, for example? If the authors controlled feet within the study time period, would the sentence not needed to be stated “… we review prosthetic feet ALSO in Africa.”? Or did the authors not review the feet at the study place, but only in Africa? Please clarify and rephrase – thank you. These clarifications/details seem to be better placed within the Materials and Methods Chapter.

Lines 287-311, in particular lines 311-317. Because the authors use an experimental setup, variables are usually considered as ‘independent’ or ‘dependent’. Please clarify what variables (independent, dependent) were considered. All variables the authors used to classify the different feet need be listed together with the rationale for the decision to include them into the regression model, including their order.

Ok – are the variables listed on line 314 the once considered as independent variables to classify a prosthetic foot as ‘reusable’ or ‘not reusable’? If so, why? In other words, why do the authors consider the ‘brand’ to be more important than the condition of the foot itself? Or is it the case because only those feet that passed the visual test of being 'usable' were included into the regression model? Up to this point, this reviewer understood the aim of the study as being to develop a set of criteria to classify a prosthetic foot as reusable or not. If a visual pre-selection needs to be carried out first, how good/robust is the regression model that follows? Please clarify and add this clarification to the text – thank you.

Lines 296-298. What variables (the authors call them ‘flaws’, if understood correctly) did the authors consider during the usability tests? How were these tests carried out? What happened with the feet the authors did consider as ‘not usable’: where they removed from the total sample of 366 feet (see below remarks to line 319)? For illustration: assuming the authors used for their visual check a variable called ‘cracks within the cosmetic’: did the authors classify a foot as still usable when only surface cracks were available, or did they exclude any foot with a crack in its shell? What were the criteria to classify a SACH foot as ‘usable’? More detailed information about the entire method for the visual checks and the resulting classification needs to be stated.

When did the authors add any of this variable into the regression model and they give some of the variables a weighting, i.e. were some of the variables considered more important than others, and if so, why? Please add this information and make a reference to Table 2 or better, create a new Table or flowchart showing the authors thoughts and decision process including the variables used upon which they based their decision to classify a foot as ‘usable’ or ‘not usable’. Clarification on this matter will strengthen the work as it helps the reader to better understand the authors’ rationale – thank you!

Line 319. Please start the results section with “A total of 366 feet where analyzed, 196 left and 170 right feet…”

Line 320. Please add “… and A brand could be identified for… ” – thank you.

Lines 320-322. Based on the information given in Table 1, there were 12 brands identified as categories plus one category with feet unknown to the authors. Because ‘unknown’ is not a brand, the sentence needs to be rephrased – thank you.

Lines 353-357. These sentences seem to be missing some text, at least, they do not make sense to this reviewer. In lines 353-355 the authors state that the feet of Trulife and Ossur performed worst. Then in the following lines the authors state that they are (nevertheless??) considered as appropriate for donation. Please clarify – thank you.

Table 4. Please explain/add, either in the corresponding text (lines 350 and subsequently) how the negative signs have to be read. Why has the measurement made against ‘BioQuest’ and not ‘Janton’ and how do the authors explain the difference in the coefficient between these two feet? Both feet were represented with n=1, why is there a difference? Please explain and add the clarification into the text within the Discussion section – thank you.

Figure 2. Please add to Fig. 2, a, b, and c, as done in Fig. 1. This assists in clarifying matters. Please add this clarification into the text: line 364 = Figure 2a; line 378: delete (Figure 2) and add after ‘NCRPPD’ (Figure 2b); line 379: add (Figure 2c) after ‘K4C’.

Line 388. Add at the end of the sentence ‘(Figure 3)’.

Line 395. Please expand this sentence like or similar as proposed “…can be a burden to the recipient LMIC [31, 39,40], as indicated by Marks et al (2019 – Please check PLOS rules!!):” and then have the quotation followed. This will connect the quotation with the text and makes it easier to read.

Line 469. Please check this sentence – the word ‘design’ seems to be twice stated. If this is correct, consider rephrasing as the sentence reads strange, thank you.

Checklist questions:

• Question (1): Please add example of ‘completeness’ of a prosthetic foot, as you did for Question 2.

• Question (3): Add examples of what the authors consider ‘compliant’: forefoot, heel, middle section? All of these, only one? Usable for light persons, like children if only one part of the foot is too compliant? If so, which one do the authors consider as the most important variable for a foot to be still considered ‘usable’?

Line 529. Word missing: “..cost of what” was the biggest barrier? Please complete.

Line 533. Please consider replacing ‘in this way’ with ‘Therefore’ or similar that would connect clearer the content of the previous paragraph with this new one.

Line 544. Typos: ‘reduce’ instead of ‘reduces’, ‘limit’ instead of ‘limits’.

Line 567. Stop the sentence after ‘repair of equipment’ and continue with a new sentence starting, for example with “Hamner et al (please check PLOS rules!!) point out that … and than add the quotation.

Line 570. Please delete ‘etc.’ This should not be used in a text as it lefts the reader wonder what else – in this case – could have had an influence. Instead write ‘for example’ and list the three most missing points that were not considered.

Line 620. Keep the number correct: the authors tested 306 feet. The number speaks for itself, no need to bolster it. To this reviewer bolstering looks bad, stay with the figures.

Line 622. Replace ‘are’ with ‘were’, as this was the case for the authors' sample. Samples of other authors might vary.

**Do you want your identity to be public for this peer review?** For information about this choice, including consent withdrawal, please see our Privacy Policy..

Reviewer #1: No

Reviewer #2: No

Reviewer #3: **Yes:** Margrit R. Meier, PhDMargrit R. Meier, PhDMargrit R. Meier, PhDMargrit R. Meier, PhD

---

## [Decision Letter · Decision Letter 1]

10 Mar 2026

PGPH-D-25-02840R1

Proposed standards for prosthetic foot reuse and considerations for donation of used prosthetic feet to low-and middle-income countries

Dear Dr. Berthaume,

Thank you for submitting your manuscript to PLOS Global Public Health. After careful consideration, we feel that it has merit but does not fully meet PLOS Global Public Health’s publication criteria as it currently stands. Therefore, we invite you to submit a revised version of the manuscript that addresses the points raised during the review process.

The manuscript has been evaluated by one reviewer, and the comments are provided below. Overall, the reviewer thanks you for submitting the revised version of the manuscript. The authors have carefully addressed the previous comments, and the manuscript has improved substantially as a result. The revisions have strengthened the presentation and clarity of the work. Only a few minor comments remain.

Could you please revise the manuscript to address the remaining points raised by the reviewer?

We look forward to receiving your revised manuscript.

Kind regards,

Katrien G. Janin, PhD

Staff Editor

Journal Requirements:

Additional Editor Comments (if provided):

Reviewers' comments:

Reviewer's Responses to Questions

**Comments to the Author**

Reviewer #1: All comments have been addressed

Reviewer #2: All comments have been addressed

Reviewer #3: All comments have been addressed

publication criteria? Is the manuscript technically sound, and do the data support the conclusions? The manuscript must describe methodologically and ethically rigorous research with conclusions that are appropriately drawn based on the data presented.? Is the manuscript technically sound, and do the data support the conclusions? The manuscript must describe methodologically and ethically rigorous research with conclusions that are appropriately drawn based on the data presented.

Reviewer #1: Yes

Reviewer #2: Yes

Reviewer #3: Yes

3. Has the statistical analysis been performed appropriately and rigorously?

Reviewer #1: N/A

Reviewer #2: Yes

Reviewer #3: Yes

4. Have the authors made all data underlying the findings in their manuscript fully available (please refer to the Data Availability Statement at the start of the manuscript PDF file)?

The PLOS Data policy requires authors to make all data underlying the findings described in their manuscript fully available without restriction, with rare exception. The data should be provided as part of the manuscript or its supporting information, or deposited to a public repository. For example, in addition to summary statistics, the data points behind means, medians and variance measures should be available. If there are restrictions on publicly sharing data—e.g. participant privacy or use of data from a third party—those must be specified.requires authors to make all data underlying the findings described in their manuscript fully available without restriction, with rare exception. The data should be provided as part of the manuscript or its supporting information, or deposited to a public repository. For example, in addition to summary statistics, the data points behind means, medians and variance measures should be available. If there are restrictions on publicly sharing data—e.g. participant privacy or use of data from a third party—those must be specified.

Reviewer #1: Yes

Reviewer #2: Yes

Reviewer #3: Yes

5. Is the manuscript presented in an intelligible fashion and written in standard English?

Reviewer #1: Yes

Reviewer #2: Yes

Reviewer #3: Yes

Reviewer #1: Authors have incorporated all the comments.

Reviewer #2: All of the corrections have been made.

Reviewer #3: GENERAL

Congratulations to the authors for having completed the review. The amendments added to the manuscript have a large positive effect and strengthened it substantially. This reviewer also welcomes the clarifications added to the title: it captures in my point of view the essence of the manuscript. Excellent!

Thank you also for the big compliment I, as reviewer 3, have received. This was unexpected and much appreciated indeed!

Also appreciated is the authors’ response to my statistical question I had. Only some minor issues remain – please verify and adapt accordingly, thank you!

DETAILED

Line 46: If word count allows, please spell out LMICs, thank you.

Line 47: Missing full stop.

Line 93: Thank you for the explanation given re ‘prosthetics’ and the changes you made throughout the manuscript. One seem to have been overlooked: please change ‘prosthetics’ to either ‘prostheses’, ‘prosthetic components’, ‘prosthetic provision’, ‘assistive devices’ or similar as it assists in clarifying matters – than you.

Line 146: Please change ‘prosthetics’ to ‘prostheses’.

Lines 235-38: Please change ‘prosthetics’ to ‘prostheses’ in all cases, as – according to this reviewers understanding, the authors are referring to an entire artificial device that replaces a limb. However, if you also refer to hearing aids or orthoses, then it is recommended to use the wider term ‘assistive devices’.

Line 249: Please change ‘prosthetics’ to ‘prostheses’, as the authors refer to an entire prosthesis. If this is not the case, then delete the word ‘prosthetics’ as it would repeat the word ‘prosthetic components” – thank you!

Line 303: a word seems to be missing: “(...) feet already AT/IN? prosthetic and orthotic centers (…).

Line 482: Space missing between the words ‘permitted’ and ‘Ethical’.

Line 508. Please delete the lonely full stop and adjust spacing – thank you.

Line 542: please replace ‘prosthetics’ with either ‘prosthetic devices and componentry’ or ‘assistive devices and componentry’.

Line 593: Please rephrase the start of this sentence to “Prosthetic feet can last long in LMICs (refs as stated) as they are overdesigned for their intended length of use (…) – thank you.

Line 644: perhaps better “(…), device design is THE focus, (…)?

Line 667: please delete ‘a’ so that it reads: “(…) proposed set of standards in (…)” – thank you.

Lines 694-5: please add ‘for example’ after ‘is’ and replace ‘etc.,’ with ‘or other variables’ so that the line reads: “(…) usability is, for example, related to material, prosthetic component use, P&O centre policies/magnitude of use prior to donation, or other variables.” Thank you!

**Do you want your identity to be public for this peer review?** For information about this choice, including consent withdrawal, please see our Privacy Policy..

Reviewer #1: No

Reviewer #2: No

Reviewer #3: **Yes:** Margrit R. Meier, PhDMargrit R. Meier, PhDMargrit R. Meier, PhDMargrit R. Meier, PhD

---

## [Decision Letter · Decision Letter 2]

23 Mar 2026

Proposed standards for prosthetic foot reuse and considerations for donation of used prosthetic feet to low-and middle-income countries

PGPH-D-25-02840R2

Dear Dr. Berthaume,

We are pleased to inform you that your manuscript 'Proposed standards for prosthetic foot reuse and considerations for donation of used prosthetic feet to low-and middle-income countries' has been provisionally accepted for publication in PLOS Global Public Health.

Best regards,

Julia Robinson

Executive Editor

Reviewer Comments (if any, and for reference):

Reviewer's Responses to Questions

**Comments to the Author**

Reviewer #2: All comments have been addressed

Reviewer #3: All comments have been addressed

publication criteria? Is the manuscript technically sound, and do the data support the conclusions? The manuscript must describe methodologically and ethically rigorous research with conclusions that are appropriately drawn based on the data presented.? Is the manuscript technically sound, and do the data support the conclusions? The manuscript must describe methodologically and ethically rigorous research with conclusions that are appropriately drawn based on the data presented.

Reviewer #2: Yes

Reviewer #3: Yes

3. Has the statistical analysis been performed appropriately and rigorously?

Reviewer #2: Yes

Reviewer #3: Yes

4. Have the authors made all data underlying the findings in their manuscript fully available (please refer to the Data Availability Statement at the start of the manuscript PDF file)?

The PLOS Data policy requires authors to make all data underlying the findings described in their manuscript fully available without restriction, with rare exception. The data should be provided as part of the manuscript or its supporting information, or deposited to a public repository. For example, in addition to summary statistics, the data points behind means, medians and variance measures should be available. If there are restrictions on publicly sharing data—e.g. participant privacy or use of data from a third party—those must be specified.requires authors to make all data underlying the findings described in their manuscript fully available without restriction, with rare exception. The data should be provided as part of the manuscript or its supporting information, or deposited to a public repository. For example, in addition to summary statistics, the data points behind means, medians and variance measures should be available. If there are restrictions on publicly sharing data—e.g. participant privacy or use of data from a third party—those must be specified.

Reviewer #2: Yes

Reviewer #3: Yes

5. Is the manuscript presented in an intelligible fashion and written in standard English?

Reviewer #2: Yes

Reviewer #3: Yes

Reviewer #2: The authors have addressed all the issues in the manuscript.

Reviewer #3: Thank you for addressing the remaining issues completely. This is most appreciated as the manuscript's content has gained once more on clarity. Thank you again.

**Do you want your identity to be public for this peer review?** For information about this choice, including consent withdrawal, please see our Privacy Policy..

Reviewer #2: No

Reviewer #3: No
